# Development and Validation of Micro-Azocasein Assay for Quantifying Bromelain

**DOI:** 10.3390/mps7020025

**Published:** 2024-03-15

**Authors:** Krishna Pillai, Javed Akhter, Ahmed H. Mekkawy, Sarah J. Valle, David L. Morris

**Affiliations:** 1Mucpharm Pty Ltd., Sydney, NSW 2217, Australia; panthera6444@yahoo.com.au (K.P.); javed.akhter@health.nsw.gov.au (J.A.); ahmed.mekkawy@unsw.edu.au (A.H.M.); sarah.valle@health.nsw.gov.au (S.J.V.); 2Department of Surgery, St. George Hospital, Sydney, NSW 2217, Australia; 3St. George & Sutherland Clinical School, University of New South Wales, Sydney, NSW 2217, Australia; 4Intensive Care Unit, St. George Hospital, Sydney, NSW 2217, Australia

**Keywords:** azocasein, bromelain, enzymes, N-acetylcysteine, proteolytic activity, validation

## Abstract

The proteolytic activity of enzymes may be evaluated by a colorimetric method with azocasein. Hence, we developed a micro-assay to quantify bromelain using azocasein. A total of 250 µL of 1.0% azocasein in dH_2_O was added to 250 µL of test solution, vortexed and incubated at ambient room temperature/30 min. The reaction was terminated with 1500 µL of 5% trichloroacetic acid, vortexed and centrifuged. A total of 150 µL of 0.5M NaOH was added to 150 µL of supernatant in triplicates, and absorbance was recorded at 410 nm. The linearity of the calibration curve was tested with 200–800 µg/mL serial dilutions. The detection limit, precision, accuracy, and robustness were tested along with the substrate enzyme reaction time and solvent matrix effect. Good linearity was seen with serially diluted 200 µg/mL bromelain. The limit of quantification and limit of detection were 5.412 and 16.4 µg/mL, respectively. Intra-day and inter-day analyses showed a relative standard deviation below 2.0%. The assay was robust when tested over 400–450 nm wavelengths. The assays performed using dH_2_O or PBS diluents indicated a higher sensitivity in dH_2_O. The proteolytic activity of bromelain was enhanced with L-cysteine or N-acetylcysteine. Hence, this micro-azocasein assay is reliable for quantifying bromelain.

## 1. Introduction

Several assays have been developed for quantifying proteolytic enzymes using chemicals that are hydrolyzed by the enzyme [1,2,3,4]. The azocasein assay is one such assay, where an azo (coloring) component is linked to casein (protein), and when the latter is hydrolyzed by the enzyme, the azo dye is released. The azo dye (yellowish in color) is then detected using UV spectroscopy at 410–440 nm [1]. Using this assay, several other proteolytic enzymes may be quantified based on their proteolytic potential.

The azocasein assay may be performed using large volumes of solution in cuvettes, and the azo dye released after proteolysis can be detected using UV spectroscopy. In the micro-assay, a small volume (µL) of reagents is used, and the final detection of the released dye can be carried out using very small volumes varying from 50–100 µL, which are decanted into a microwell. One of the major advantages of the micro-assay is that several different reactions can be conveniently carried out using this assay, with an accuracy that is equivalent to a macro-standard assay. Hence, this micro-assay is not only economical, since a small number and volume of reagents are used, but furthermore, several different enzymes and different reactions can be carried out simultaneously and evaluated both economically and conveniently.

Bromelain is a proteolytic enzyme that is widely used in industries for the hydrolysis of chitinous materials and complex carbohydrates [5]. Bromelain possesses a range of properties, including fibrinolytic, anti-edematous, anti-thrombotic, and anti-inflammatory activities [6]. In the pharmaceutical industries, it is used in wound debridement (Nemourid) [7], and more recently, it has been evaluated for the treatment of rare cancer known as pseudomyxoma peritonei (PMP) where cancerous cells secrete mucinous material in the peritoneal cavity [8]. With its proteolytic properties, bromelain acts as a mucolytic by acting on the mucinous glycoprotein peptide and glycosidic linkages, whereby depolymerization of polymeric mucin occurs [9]. It is currently combined with N-acetylcysteine, an antioxidant that effectively reduces the disulfide linkages found in between the polymeric glycoprotein of mucin. Hence, together these two agents are effective mucolytics, besides also having anti-cancer properties [10].

Therefore, in the current work, we demonstrate the methodology of setting up a micro-assay to detect the proteolytic activity of bromelain for quantification and validation using the ICH Harmonised Guideline: Validation of Analytical Procedures Q2(R2) [11], with the effect of other parameters on the assay, such as the reaction time, solvent matrix effect and effect of antioxidants. We also show that the present micro-assay is more sensitive than the earlier assay set up by Coelho et al. [1].

## 2. Experimental Design

### 2.1. Materials

Bromelain (Enzybel, Villers-Le-Bouillet, Belgium).Azocasein (Sigma Aldrich, St. Louis, MO, USA, cat. A2765).Cysteamine (Sigma Aldrich, St. Louis, MO, USA, cat. 30070).Dithiobutylamine (DTBA) (Sigma Aldrich, St. Louis, MO, USA, cat. 774405).Dithiothreitol (DTT) (Sigma Aldrich, St. Louis, MO, USA, cat. 43815).N-acetylcysteine (NAC) (Sigma Aldrich, St. Louis, MO, USA, cat. A7250).L-cysteine (Sigma Aldrich, St. Louis, MO, USA, C7352).All other reagents were purchased from Sigma Aldrich, (St. Louis, MO, USA).

### 2.2. Equipment

UV spectrometer (Shimadzu, Kyoto, Japan, cat. UV-1800).

## 3. Procedure

### 3.1. Calibration Curves

Prepare a fresh stock solution of bromelain 10 mg/mL in distilled water (dH_2_O).For standard curve preparation, prepare a 200 µg/mL bromelain solution in dH_2_O. Then, carry out a serial dilution starting from 200 µg/mL and below in dH_2_O. Other standard curves that were serially diluted, such as 800, 600, and 400, were also investigated to arrive at linearity.Prepare 1% azocasein (*w*/*v*) in dH_2_O. Weigh the required quantity of azocasein and dissolve it in dH_2_0 by vigorous vortex mixing. For a volume of 50 mL, 1.0 mL of ethanol (absolute) is added to the solids before adding the amount of dH_2_O.0.5 M sodium hydroxide is prepared by dissolving the required amount in dH_2_O.Add 250 µL of azocasein solution to 250 µL of each bromelain concentration in a polypropylene tube at room temperature. The control contains only dH_2_O + 1% azocasein.Vortex mix the tubes and place them on an agitator shaker at ambient room temperature (23 °C) for 1 h.Add 1500 µL of 5% (*w*/*v*) trichloroacetic acid (TCA) that was previously mixed and centrifuged at 3000 rpm for 7 min. TCA is prepared by dissolving the required amount in dH_2_O.Pipette 150 µL of the clear yellow supernatant into the microwell of a 96-microwell plate from each dilution in triplicates.Add 150 µL of 0.5 Sodium hydroxide to each well to stop the reaction.Read the absorbance at 410 nm using a UV spectrophotometer.For unknown samples, similarly, 250 µL of each sample is treated as in steps 5–10. The concentration of unknown samples is calculated from the bromelain standard curve. The unknown bromelain sample should be suitably diluted to fall within the range of 200 µg/mL and below, treated with azocasein, and the absorbance can be measured to quantify the bromelain content.

### 3.2. Linearity of Calibration Curve

Serially diluted samples of bromelain with values of 600, 400, 300 and 200 were treated with the azocasein assay and evaluated for linearity of the calibration curves. The correlation coefficient (R^2^) value was used to determine the linearity of the graph.

### 3.3. Detection Limit

Three series of calibration curves were plotted by serial dilutions (0–200 µg/mL) in dH_2_O using a stock solution of bromelain. UV absorption was measured in triplicates at 410 nm. The limits of detection (LOD) and the limit of quantification (LOQ) were evaluated directly from the calibration curve, as stipulated by the ICH guidelines [11].
LOD = 3.3 σ/S and LOQ = 10.0 σ/S;
where σ = mean of the standard deviation of intercept; S = mean of the slopes of the calibration curves.

### 3.4. Precision

Intra-day precision was evaluated by measuring the concentration of three different samples of the same concentration (25.0 µg/mL) in triplicates under the same experimental conditions on the same day following the sample preparations described earlier.

Inter-day precision was evaluated by measuring the concentrations of three different samples of the same concentrations (25.0 µg/mL) in triplicates under the same experimental conditions on two different days following the sample preparations described earlier.

### 3.5. Accuracy

Accuracy was evaluated by assaying, in triplicates, samples of known concentrations of bromelain with the addition of different concentrations of bromelain (i.e., adding 3.0, 4.0, and 9.0 µg/mL to 25 µg/mL of known concentrations of bromelain) using samples prepared in dH_2_O as described earlier.

### 3.6. Robustness

The robustness of the assay was evaluated by analysis of sample solutions in comparison to standard (25.0 µg/mL) at wavelengths of 400–450 nm.

### 3.7. Reaction Time

Two sets of bromelain dilutions from 200 µg/mL and below were prepared in duplicates, and the azocasein assay was performed as outlined earlier; however, the reaction of the first was terminated at 30 min and the second at 60 min using 5% trichloroacetic acid.

### 3.8. Solvent Matrix Effect

Calibration curves (0–200 µg/mL) of bromelain were prepared in either dH_2_O or in Phosphate buffer Saline (PBS) in triplicates. Their slopes were compared initially with a subsequent comparison of LOD and LOQ values.

### 3.9. Effects of N-Acetylcysteine and Other Antioxidants on Calibration Curve

Bromelain 200 µg/mL was prepared with 2% of various antioxidants, a serial dilution was carried out in PBS, and the calibration curves were plotted. The slopes of the curves were compared to determine the enhancement of activity against the standard bromelain calibration curve.

### 3.10. Statistical Analysis

Data were reported as the mean ± SD. Qualitative variables were compared using Student’s *t*-test. Differences were considered statistically significant when *p* < 0.05.

## 4. Results

### 4.1. Linearity of Calibration Curves

The different plots for detecting the linearity of the calibration curve (Figure 1) indicated that at 200 µg/mL, bromelain serially diluted down showed good linearity with a correlation coefficient (R^2^) value of 0.992 (Figure 1D). The detection range was between 3.125 and 200 µg/mL. 

### 4.2. Limits of Detection (LOD) and Limits of Quantification (LOQ)

Three series of calibration curves were plotted by serial dilutions (Figure 2). Both parameters, limits of detection (LOD) and limits of quantification (LOQ), were determined from the graph using principles stipulated in the ICH Harmonized Guidelines [11].
LOD = 3.3 (σ)/S
where σ = the standard deviation of the response and S = the slope of the calibration curve
LOD = 3.3 (0.00338)/0.002061  = 5.412 µg/mL
LOQ = 10 (σ)/S          = 10 (0.00338)/0.002061      = 16.4 µg/mL

Although the calculated LOQ was 16.4 µg/mL, practically speaking, low levels that are even below 5.0 µg/mL can also be detected, although the accuracy may vary (Figure 2).

### 4.3. Precision (Intra-Day and Inter-Day Measurement)

The intra-day and inter-day measurements, as shown in Table 1, indicate that the relative standard deviation (RSD) fell below 2.0%, indicating that variance is very small and that analysis using the current method may be performed with good precision.

### 4.4. Accuracy

A further accuracy analysis, shown in Table 2, indicates that it is close to 100% (mean recovery = 99.56 ± 0.17).

### 4.5. Robustness

The reliability of the assay was assessed by evaluating the robustness of the assay at different wavelengths, as shown in Table 3. It indicated that almost 100% of the bromelain content was evaluated (Table 3).

### 4.6. Enzyme-Substrate Reaction Time (OD in Relation to Proteolysis)

Two bromelain serial dilutions (0–200 µg/mL) were prepared, and the azocasein reactions were stopped at either 30 or 60 min (Figure 3). A comparison of the 1/slopes of the two calibration curves, 30 min vs. 60 min, shows that there is a much better response at 60 min (color as shown by the OD 400 nm).

733.9/1228 (100) = 59.76; 100 − 59.76 = 40.24%; hence, there is a difference of 40.24% in the signal (Figure 3).

### 4.7. Solvent Matrix Effect

Figure 4A: Mean slope ± S.D. = 0.001665 ± 3.5122 × 10^−5^Figure 4B: Mean slope ± S.D. = 0.002059 ± 7.4087 × 10^−6^When comparing the slopes, 0.002059/0.001665 (100) = 123.66 − 100 = 23.66%.A difference of 23.66% is observed.

**Figure 4 mps-07-00025-f004:**
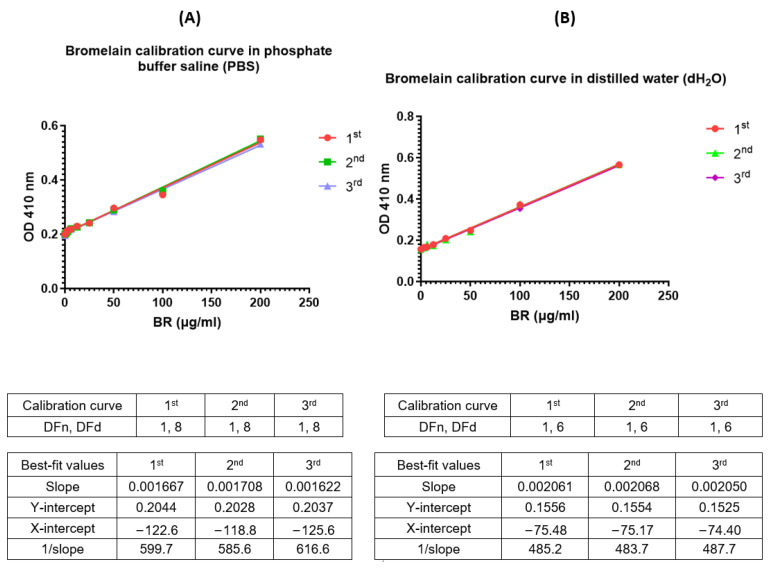
Three identical linear plots of bromelain dilution (200 µg/ml) in Phosphate Buffer Saline (PBS) (**A**); and distilled water (dH_2_O) (**B**).

Based on the LOD and LOQ values (Table 4), a slightly lower quantity of bromelain can be detected in dH_2_O as compared to PBS, and the same applies to the LOQ values. Hence, the bromelain standard curve prepared in dH_2_O is slightly more sensitive.

### 4.8. Effects of N-Acetylcysteine and Other Antioxidants on the Calibration Curve

Amongst the various agents added, the proteolytic activity of bromelain with only L-cysteine and NAC was enhanced when compared to bromelain without any antioxidant addition (Figure 5).

Amongst the various agents added, the proteolytic activity of bromelain with only L-cysteine and NAC was enhanced when compared to bromelain without any antioxidant addition (Figure 5).
L-cysteine = (1/slope of L-cysteine)/(1/slope of No additive) × 100        = 469.7/576.9 (100) = 81.4; 100 − 81.4 = 18.6% enhancement
N-acetylcysteine = (1/slope of N-acetylcysteine)/(1/slope of No additive) × 100        = 553.3/576.9 (100) = 95.9; 100 − 95.9 = 4.1% enhancement

The addition of NAC only very slightly enhanced the proteolytic activity of bromelain at pH 7.4.

All the other additives, such as Cysteamine, DTT, Ascorbic acid, and DTBA, showed an inhibitory action on the proteolytic activity of bromelain at pH 7.4.

Comparison to the proteolytic activity of bromelain with no additives using 1/slope values:Cysteamine = (1/slope of Cysteamine)/(1/slope of No additive) × 100       = 6985/576.9 (100) = 1210; 100 − 1210 = −1110 depression
DTT = (1/slope of DTT)/(1/slope of No additive) × 100         = 1301/576.9 (100) = 225.5; 100 − 225.5 = −125.5% depression
Ascorbic acid = (1/slope of Ascorbic acid)/(1/slope of No additive) × 100         = 1410/576.9 (100) = 244.4; 100 − 244.4 = −144.4% depression
DTBA = (1/slope of DTBA)/(1/slope of No additive) × 100         = 1318/576.9 (100) = 228.5; 100 − 228.5= −128.5% depression

Amongst the antioxidants evaluated, only N-acetylcysteine and L-cysteine enhanced the activity of bromelain at pH 7.4, whilst all others showed a depression of proteolytic activity as assessed using the azocasein assay (Table 5). 

## 5. Discussion

The aim of the current work is twofold, i.e., to set up a micro azocasein assay and to validate the assay for quantifying bromelain through its proteolytic action on a substrate known as azocasein. The proteolysis (hydrolysis) of azocasein releases a yellowish azo dye that is detected at UV ranging from 400–450 nm. The intensity of color produced by the azo dye that is measured by UV spectroscopy can be used to quantify bromelain in a solution in the µg range.

Initially, serially diluted samples of bromelain with values of 800, 600, 400, etc. and below showed that using a 200 µg/mL bromelain solution that is serially diluted gives a good linear graph of absorbance vs. concentration. The correlation coefficient was 0.992, indicating good linearity, with the representative linear equation being:y = 0.001681 × x + 0.1463 (*p* ≤ 0.0001);
which is calculated by the least square method.

The LOQ was found to be 16.4 µg/mL, whilst the LOD was 5.412 µg/mL. Although the LOQ calculated was 16.4 µg/mL, values as low as 5.0 µg/mL or even lower may be quantified.

Further, the intra-day and inter-day analysis showed that the relative standard deviation (R.S.D) values that were obtained by the designated method were found to be below 2.0%, indicating that variance is rather small and that analysis may from time to time be performed with a good accuracy that is close to 100% (mean recovery = 99.56 ± 0.17).

The reliability of the assay was tested by the assay robustness, with a measurement of the absorbance of the chromogen (azo dye) at different wavelengths (400–450 nm), which indicated almost 100% of the bromelain content evaluated. Hence, the azocasein assay is a reliable method for quantifying bromelain.

In addition, to validate the assay, we also examined certain other parameters in the assay, such as enzyme hydrolysis of casein (reaction time) at 30 vs. 60 min. The results indicated that terminating the reaction at 60 min as compared to 30 min provided a better response, as indicated by the slope of the graphs and the absorbance (OD), with a 40.24% increase in the signal. This is in agreement with other researchers who have monitored the time-dependent hydrolysis of casein [1].

Although bromelain can be quantified in either dH_2_O or in phosphate buffer saline, we examined the slope of the graphs in two media with similar dilution, and we found that there is a difference of 23% in favor of dH_2_O, indicating a steeper slope with dH_2_O. The zero absorbance was also relatively higher for PBS, although both media were at pH 7.4. When further evaluating the LOD and LOQ values, there is the indication that the assay is slightly more sensitive in dH_2_O, their respective values being LOD = 5.39 µg/mL and LOQ = 16.348 µg/mL for dH_2_O, as compared to PBS with LOD = 5.86 µg/mL and LOQ = 17.752 µg/mL.

Finally, we examined the effect of antioxidants on the proteolytic activity of bromelain, since, in our mucolytic formulation (BromAc), bromelain at 600 µg/mL is found in a solution of 2% N-acetylcysteine (20 mg/mL) at pH 7.0. A series of antioxidants were used for comparison to determine their effect on bromelain. Comparing the enhancement of proteolytic activity to standard bromelain without additives, only L-cysteine with 18.7% and N-acetylcysteine with 4% were on the list of enhancers, while all others showed a depression. Strong antioxidants such as DTT and DTBA only showed a depression, indicating that the pH of the media at 7.4 may have influenced this observation. On the other hand, the concentration of the antioxidants used may have affected the bromelain. Earlier studies with cysteamine have indicated that a combination of 2.0% cysteamine with 12.5 µg/mL bromelain showed a maximal proteolytic increase in activity of about 48%, whilst 1.0% NAC with 25 µg/mL bromelain gave an 87% enhancement, as compared to controls (with no antioxidants) (unpublished data). Various chemical agents investigated for their effect on bromelain indicated different degrees of proteolytic enhancement on bromelain extracted from different parts of the plant or unripe fruit, as compared to ripe fruits [11].

Bromelain has 222 amino acids, of which three are cysteine residues that carry an S-H group [12]. This group can get oxidized to form the disulfide linkages, resulting in the loss of proteolytic activity. Hence, the addition of cysteines and other reducing agents may restore the native state of the cysteine residues and thus may bring back the proteolytic activity at the catalytic site, thus showing an increase in activity. However, antioxidants are pH-sensitive, and hence a pH of 7.4 may not be favorable for all antioxidants [13], which leads to this observation.

Although other methods such as HPLC may provide a more sensitive quantification of bromelain, the current method may serve as a convenient and fast method of quantification where a high sensitivity for detecting nanograms of the enzyme is not required. Our earlier studies have also indicated that the present assay is not suitable for quantifying bromelain in patient serum owing to interference by the presence of endogenous proteolytic enzymes as well as other components that may react with bromelain. In addition, the bromelain present in blood occurs in the nanogram range, which the micro-azocasein assay is unable to detect owing to a lack of sensitivity. Hence, the micro-azocasein assay is a reliable and robust assay that can be routinely used in the laboratory in an economic way, since it involves neither a very laborious set-up nor expensive reagents.

Finally, although earlier work by Coelho et al. [1] has developed a miniature essay for quantifying bromelain, it neither uses the present micro level of reagents nor is their assay as sensitive. In comparison to the present assay, their LOD and LOQ was 72 and 494 µg/mL and 5.412 and 16.4 µg/mL, respectively, indicating that the current assay is several times more sensitive and useful, with much lower-level detection limits being required.

## 6. Conclusions

We conclude that the micro-azocasein assay is a convenient, economical, and versatile assay, since it may be applicable to assessing the proteolytic activity of different enzymes, and in particular cysteine proteases, with a sensitivity that exceeds an earlier assay that was published by another group.

## Figures and Tables

**Figure 1 mps-07-00025-f001:**
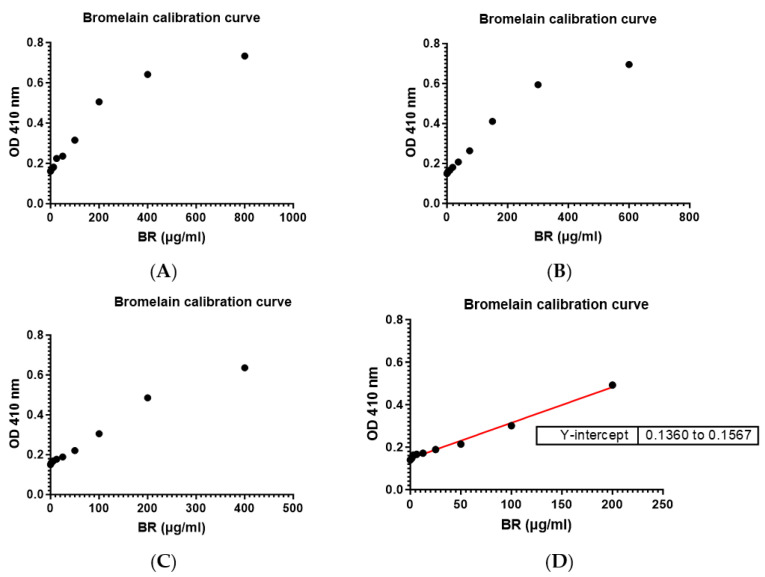
The linear bromelain calibration curve was generated using different concentrations of Bromelain (i.e., serial dilutions of Bromelain at 800 (**A**); 600 (**B**); 400 (**C**); and 200 (**D**) µg/mL) to demonstrate linearity.

**Figure 2 mps-07-00025-f002:**
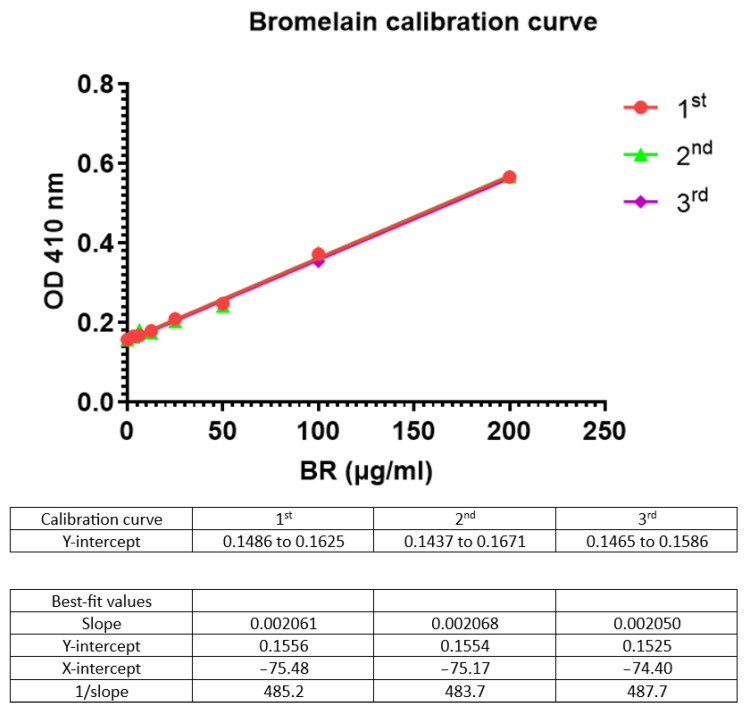
Three series of bromelain linear calibration curves (1st, 2nd, and 3rd) were plotted by serial dilutions (0–200 µg/mL). The graph shows that the slopes are almost identical: 0.002061, 0.002068, and 0.002050.

**Figure 3 mps-07-00025-f003:**
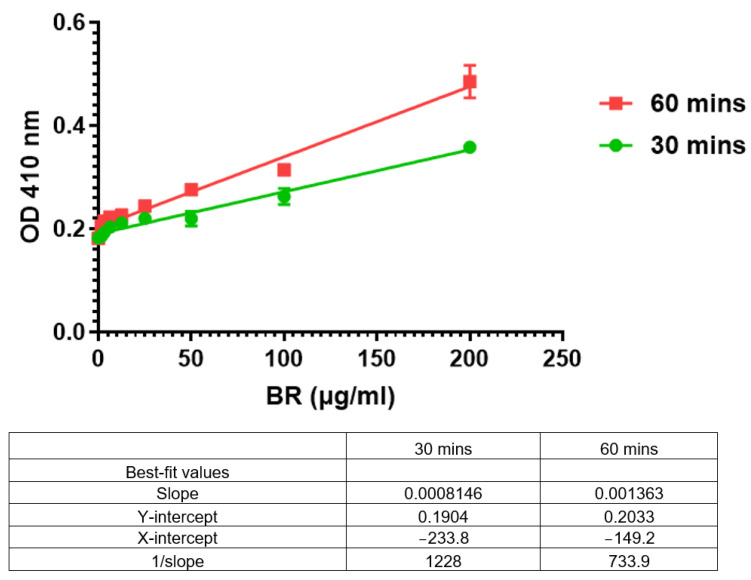
The effect of enzyme-substrate reaction time on the bromelain calibration curve. At 60 min, the linear plot has a higher slope value compared to the 30 min linear curve.

**Figure 5 mps-07-00025-f005:**
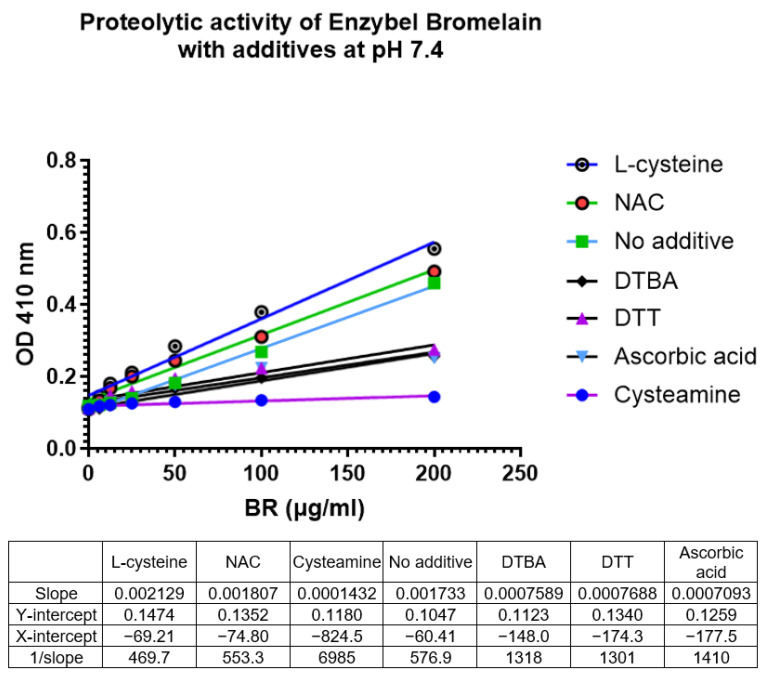
Different calibration curves for bromelain with the addition of various antioxidants. L-cysteine has the highest slope value compared to the rest of the additives. DTBA: Dithiobutylamine; DTT: Dithiothreitol; NAC: N-acetylcysteine.

**Table 1 mps-07-00025-t001:** Intra-day (repeatability) and inter-day precision (intermediate precision) of the method using a theoretical concentration of 25 µg/mL. Data expressed as mean ± SD. RSD: Relative Standard Deviation.

Precision	Measured Concentration (µg/mL)	Mean Recovery (%)	RSD (%)
Intra-day (n = 3)	25.886 ± 0.002	103	0.37
Inter-day Day1 (n = 3)	25.926 ± 0.163	103.7	0.6
Inter-day Day2 (n = 3)	25.093 ± 0.133	100.37	0.53
Inter-day Day1 & 2 (n = 6)	25.509 ± 0.416	102.04	1.63

**Table 2 mps-07-00025-t002:** The table indicates the accuracy (%) of the azocasein assay. Data expressed as mean ± SD.

BR Sample (µg/mL)	Added BR (µg/mL)	BR Detected (µg/mL)	Accuracy (%)
25.085 ± 0.098	3.0	27.94 ± 0.068	99.8
25.085 ± 0.098	4.0	28.82 ± 0.0049	99.4
25.085 ± 0.098	9.0	33.898 ± 0.0121	99.5
Mean Recovery			99.56 ± 0.17

**Table 3 mps-07-00025-t003:** Robustness of bromelain azocasein method at 6 different UV wavelengths. Data expressed as mean ± SD. UV: Ultraviolet.

UV (nm)	Absorbance		Bromelain Content (%)
	Sample Solution (25 µg/mL)	Standard Solution (25 µg/mL)	
400	0.250 ± 0.001	0.252 ± 0.001	99.2
410	0.252 ± 0.0012	0.250 ± 0.0004	100.8
420	0.252 ± 0.0016	0.252 ± 0.0026	100
430	0.251 ± 0.0028	0.253 ± 0.0008	99.2
440	0.251 ± 0.0025	0.252 ± 0.0029	99.6
450	0.240 ± 0.0029	0.240 ± 0.0005	100

**Table 4 mps-07-00025-t004:** The effect of solvent matrix on limits of detection (LOD) and limits of quantification (LOQ) of the bromelain calibration curve.

Solvent	LOD (µg/mL)	LOQ (µg/mL)
Phosphate buffer Saline (PBS)	5.86	17.75
Distilled water (dH_2_O)	5.39	16.35

**Table 5 mps-07-00025-t005:** The effect of antioxidants on the proteolytic activity of bromelain at physiological pH 7.4 as assessed by comparing the slope of the graph to that of bromelain without any addition (standard).

Additives	Enhancement/Depression
L-cysteine	Enhancement
N-acetylcysteine (NAC)	Enhancement
Dithiobutylamine (DTBA)	Depression
Dithiothreitol(DTT)	Depression
Ascorbic acid	Depression
Cysteamine	Depression

## Data Availability

The datasets generated and/or analyzed during the current study are available from the corresponding author upon reasonable request.

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
