# Peer review of "Development and Validation of Micro-Azocasein Assay for Quantifying Bromelain"

_mps, 2024, doi:10.3390/mps7020025_

Round 1
Reviewer 1 Report
Comments and Suggestions for Authors
the manuscript is very interesting as it proposes the validation of a reduced analysis of protease activity using azocasein.
The validation tests were well conducted, it would only improve the quality of the graphics.
I recommend publishing the article.
Comments on the Quality of English LanguageThe English is of good quality, just a small revision to improve clarity.
Author Response
Thank you for your comments, appreciate it.
The English language has been revised.
Reviewer 2 Report
Comments and Suggestions for Authors
In this paper the Authors describe the development of the azocasein assay for bromelain quantification. The method that uses asocasein for the proteolytic activity quantitation is known for a long time – its first description was published as early as in 1947 (Charney J, Tomarelli RM. J Biol Chem. 1947;171:501-505). Its application in the bromelain activity analysis was thoroughly described several years ago (Coelho DF, et al. Biomed Res Int. 2016;2016:8409183). The present paper is very similar to that one by Coelho at al. As far as I understand, Pillai et al. aimed to develop the assay to a microscale as compared with their predecessors. As such, they should compare their results with those obtained by Coelho at al. Such a comparison is missing from the discussion. For example, the limits of detection and limits of quantification values obtained by Pillai et al. were of 5.412 μg/ml and 16.4 μg/ml, respectively. The corresponding values reported by Coelho et al. were, respectively, 72 μg/ml and 494 μg/ml. In my opinion, the Authors should include in the discussion a thorough, detailed comparison of their data with those reported by Coelho et al. That would clearly emphasized the novelty of their manuscript, which is currently weakly marked and therefore questionable. The Authors only briefly mentioned in their manuscript about the paper by Coelho et al. that in fact is of the key significance for their results.
Why the Authors did not use the two-fold bromelain dilutions, that is 800, 400, 200, 100, 50, etc? Instead, they used the following dilutions: 800, 600, 400, etc. What was the reason for that?
The Authors should use more replicates than 3 (triplicates) for the assay precision validations, including the intra-day and inter day measurements. That would increase the statistical power of the study and thus would provide more reliable idea of the assay accuracy. The minimal sample size to provide a sufficient statistical power should be calculated using an appropriate statistical software, for example the online sample size calculators are available for free in the Internet. Thus, the Authors should evaluate the minimal sample sizes for the repeatability and intermediate precision measurements and the manuscript should be resubmitted with the updated precision evaluation after an appropriate number of replicates was used.
Other comments:
- page 4, the caption to Figure 1: change “Figure 1: shows linear bromelain...” to “Figure 1: The linear bromelain...”;
- page 7, line 236: the reason for the calculations shown is unclear, please explain these calculations, show the basic formula that was used, use a separate paragraph to highlight the calculations from the text body; the same comments for the calculations shown on page 8, lines 257-259, page 9, lines 286-287 and page 9, lines 294-297;
- page 7, the caption to Figure 3: change “Figure 3: A graph shows the effect...” to “Figure 3: The effect...”.
Comments on the Quality of English LanguageThe moderate editing is required to improve the English Language Quality. Specific comments:
- page 1, line 15: change “To 150 µl of supernatant in triplicates was added 150 µl of 0.5M NaOH” to “150 µl of 0.5M NaOH was added to 150 µl of supernatant in triplicates”;
- page 1, lines 19: change “200 µg/ml bromelain serially diluted” to “serially diluted 200 µg/ml bromelain”;
- page 2, line 85: change “only contains” to “contains only”;
- page 2, line 89: change “trichloroacetic acid (TCA) is added mixed” to “trichloroacetic acid (TCA) that was previously mixed”;
- page 3, line 125: change “This was..” to “The robustness of the assay was…”;
- page 3, line 132: change “Calibration curves (0-200 µg/ml) bromelain” to “Calibration curves (0-200 µg/ml) of bromelain”.
- page 9, lines 288-289: change “The addition of NAC only enhanced the proteolytic activity of bromelain at pH 7.4 very slightly.” to “The addition of NAC only very slightly enhanced the proteolytic activity of bromelain at pH 7.4.”;
- page 10, line 303: change “Amongst the antioxidants evaluated, only N-acetylcysteine, L-cysteine” to “Amongst the antioxidants evaluated, only N-acetylcysteine and L-cysteine”;
- page 10, line 325: change “The reliability of the assay was tested by robustness of the assay” to “The reliability of the assay was tested by the assay robustness”;
- page 11, line 333: change “researcher” to “researchers”;
- page 11, line 336: change “in the two media” to “in two media”;
- page 11, line 339: change “there is indication” to “there is the indication”;
- page 11, line 349: change “only showed” to “showed only”;
- page 11, line 351: change “affected bromelain” to “affected the bromelain”;
- page 11, lines 354 and 357: change “compared” to “as compared”;
- page 11, lines 359-360: change “hence result in loss” to “that results in loss”;
- page 11, lines 360-362: change “Hence, the addition of cysteines and other reducing agents may restore the native state of the cysteine residues and hence forth the proteolytic activity” to “Hence, the addition of cysteines and other reducing agents may restore the native state of the cysteine residues and thus may bring back the proteolytic activity”;
- page 11, lines 363-364: change “and hence forth the observation” to “and hence this observation”;
- page 11, lines 368 and 372: change “nano gram” and “nano grams” to “nanogram” and “nanograms”.
Author Response
Comment 1:
In this paper, the Authors describe the development of the azocasein assay for bromelain quantification. The method that uses asocasein for the proteolytic activity quantitation is known for a long time – its first description was published as early as in 1947 (Charney J, Tomarelli RM. J Biol Chem. 1947;171:501-505). Its application in the bromelain activity analysis was thoroughly described several years ago (Coelho DF, et al. Biomed Res Int. 2016;2016:8409183). The present paper is very similar to that one by Coelho at al. As far as I understand, Pillai et al. aimed to develop the assay to a microscale as compared with their predecessors. As such, they should compare their results with those obtained by Coelho at al. Such a comparison is missing from the discussion. For example, the limits of detection and limits of quantification values obtained by Pillai et al. were of 5.412 μg/ml and 16.4 μg/ml, respectively. The corresponding values reported by Coelho et al. were, respectively, 72 μg/ml and 494 μg/ml. In my opinion, the Authors should include in the discussion a thorough, detailed comparison of their data with those reported by Coelho et al. That would clearly emphasized the novelty of their manuscript, which is currently weakly marked and therefore questionable. The Authors only briefly mentioned in their manuscript about the paper by Coelho et al. that in fact is of the key significance for their results.
Response:
Both in the abstract and in the discussion, a comparison of assay sensitivity of the earlier assay by Coelho et al to the current assay has been highlighted to show the significance of the present assay for quantifying the proteolytic activity of bromelain.
Comment 2:
Why the Authors did not use the two-fold bromelain dilutions, that is 800, 400, 200, 100, 50, etc? Instead, they used the following dilutions: 800, 600, 400, etc. What was the reason for that?
Response:
The reason for plotting individual dilution graphs for each quantity starting from 200 down etc is to show clearly how the selection of dilution concentrations affects the linearity of the graphs.
Comment 3:
The Authors should use more replicates than 3 (triplicates) for the assay precision validations, including the intra-day and inter-day measurements. That would increase the statistical power of the study and thus would provide a more reliable idea of the assay accuracy. The minimal sample size to provide sufficient statistical power should be calculated using appropriate statistical software, for example, the online sample size calculators are available for free on the Internet. Thus, the Authors should evaluate the minimal sample sizes for the repeatability and intermediate precision measurements and the manuscript should be resubmitted with the updated precision evaluation after an appropriate number of replicates was used.
Response:
Regarding using more than three triplicates, we have followed the guidelines as recommended by ICH guidelines where 3 replicates are recommended. Reference for the ICH guidelines is provided in the manuscript.
-------------------------
Other comments 1:
page 4, the caption to Figure 1: change “Figure 1: shows linear bromelain...” to “Figure 1: The linear bromelain...”;
Response:
Corrected, thank you
Other comments 2:
page 7, line 236: the reason for the calculations shown is unclear, please explain these calculations, show the basic formula that was used, and use a separate paragraph to highlight the calculations from the text body; the same comments for the calculations shown on page 8, lines 257-259, page 9, lines 286-287 and page 9, lines 294-297;
Response:
Thank you for that, all formulas are included and the text changed accordingly. Other corrections were made in page 9.
Other comments 2:
- page 7, the caption to Figure 3: change “Figure 3: A graph shows the effect...” to “Figure 3: The effect...”.
Response:
Corrected, thank you
---------------------
Comments on the Quality of English Language:
Response:
All recommended corrections were implemented, thank you
Reviewer 3 Report
Comments and Suggestions for Authors
1) The abstract does not highlight the significance of the study or the innovations.
2) Azocasein was reported to measure protease activity more than a decade ago, and the proposed method is characterised by the use of small volumes (μl) of reagents. Since the experimental principle has not changed, what is the reason for the detection capability with small sample volumes? This is not mentioned in the manuscript.
3) The logic in the 'Conclusion' section is not clear. The section stated that the proposed method is not sensitive enough and is easily interfered with when used to detect bromelain in blood. In this case, what is the appropriate scenario for using this method? It is not stated in the manuscipt. Moreover, why is it mentioned that a more sensitive ELISA method has been developed (no published data)?
4) There are some errors in the data processing. Repeated experiments under the same conditions should produce one curve instead of several as shown in the text, and each data point should be plotted with an error bar.
5) In Figure 1, the bromelain concentration corresponding to the four plots of ABCD should be given in the legend.
Comments on the Quality of English Language
Quality of English is fine.
Author Response
1) The abstract does not highlight the significance of the study or the innovations.
Response:
The significance of the study has been included in the abstract and in the discussion section.
2) Azocasein was reported to measure protease activity more than a decade ago, and the proposed method is characterised by the use of small volumes (μl) of reagents. Since the experimental principle has not changed, what is the reason for the detection capability with small sample volumes? This is not mentioned in the manuscript.
Response:
The reason for conducting the present assay is mentioned in the introduction and discussion.
3) The logic in the 'Conclusion' section is not clear. The section stated that the proposed method is not sensitive enough and is easily interfered with when used to detect bromelain in blood. In this case, what is the appropriate scenario for using this method? It is not stated in the manuscript. Moreover, why is it mentioned that a more sensitive ELISA method has been developed (no published data)?
Response:
We have corrected the conclusion section
4) There are some errors in the data processing. Repeated experiments under the same conditions should produce one curve instead of several as shown in the text, and each data point should be plotted with an error bar.
Response:
Repeated experiments have three curves, one curve for each. However, we have compared the slopes of the three curves.
5) In Figure 1, the bromelain concentration corresponding to the four plots of ABCD should be given in the legend.
Response:
The legend for Fig 1 has been corrected
Round 2
Reviewer 2 Report
Comments and Suggestions for Authors
The authors of the manuscript made corrections in accordance with my suggestions. However, previously I have missed one issue on page 8, line 43, caption to figure 4: please change "Figure 4: shows three identical.." to "Figure 4: Three identical...". Thus, the manuscript was sufficiently improved and can be published in Methods And Protocols after the last correction I have suggested herein.
Author Response
Thank you so much for revising our manuscript.
Figure 4 caption has been changed.
Reviewer 3 Report
Comments and Suggestions for Authors
The revised manuscript can be published in MPs.
Author Response
Thank you so much for reviewing the manuscript